# An Energy-Based Method for Orientation Correction of EMG Bracelet Sensors in Hand Gesture Recognition Systems

**DOI:** 10.3390/s20216327

**Published:** 2020-11-06

**Authors:** Lorena Isabel Barona López, Ángel Leonardo Valdivieso Caraguay, Victor H. Vimos, Jonathan A. Zea, Juan P. Vásconez, Marcelo Álvarez, Marco E. Benalcázar

**Affiliations:** 1Artificial Intelligence and Computer Vision Research Lab, Escuela Politécnica Nacional, Quito 170517, Ecuador; lorena.barona@epn.edu.ec (L.I.B.L.); angel.valdivieso@epn.edu.ec (Á.L.V.C.); victor.vimos@epn.edu.ec (V.H.V.); jonathan.zea@epn.edu.ec (J.A.Z.); juan.vasconez@epn.edu.ec (J.P.V.); 2Departamento de Eléctrica y Electrónica, Universidad de las Fuerzas Armadas ESPE, Sangolquí 171103, Ecuador; rmalvarez@espe.edu.ec

**Keywords:** hand gesture recognition, orientation correction, electrodes displacement, Myo armband

## Abstract

Hand gesture recognition (HGR) systems using electromyography (EMG) bracelet-type sensors are currently largely used over other HGR technologies. However, bracelets are susceptible to electrode rotation, causing a decrease in HGR performance. In this work, HGR systems with an algorithm for orientation correction are proposed. The proposed orientation correction method is based on the computation of the maximum energy channel using a synchronization gesture. Then, the channels of the EMG are rearranged in a new sequence which starts with the maximum energy channel. This new sequence of channels is used for both training and testing. After the EMG channels are rearranged, this signal passes through the following stages: pre-processing, feature extraction, classification, and post-processing. We implemented user-specific and user-general HGR models based on a common architecture which is robust to rotations of the EMG bracelet. Four experiments were performed, taking into account two different metrics which are the classification and recognition accuracy for both models implemented in this work, where each model was evaluated with and without rotation of the bracelet. The classification accuracy measures how well a model predicted which gesture is contained somewhere in a given EMG, whereas recognition accuracy measures how well a model predicted when it occurred, how long it lasted, and which gesture is contained in a given EMG. The results of the experiments (without and with orientation correction) executed show an increase in performance from 44.5% to 81.2% for classification and from 43.3% to 81.3% for recognition in user-general models, while in user-specific models, the results show an increase in performance from 39.8% to 94.9% for classification and from 38.8% to 94.2% for recognition. The results obtained in this work evidence that the proposed method for orientation correction makes the performance of an HGR robust to rotations of the EMG bracelet.

## 1. Introduction

Hand gesture recognition (HGR) systems are human–machine interfaces that are responsible for determining which gesture was performed and when it was performed [1]. Hand gestures are a common and effective type of non-verbal communication which can be learned easily through direct observation [2]. In recent years, several applications of HGRs have been proven useful. For example, these models have been applied in sign language recognition (English, Arabic, Italian) [3,4,5], in prosthesis control [6,7,8,9], in robotics [10,11], in biometric technology [12], and in gesture recognition of activities of daily living [13], among others. In the medical field, hand gesture recognition has also been applied to data visualization [14] and image manipulation during medical procedures [15,16] as well as for biomedical signal processing [17,18]. Although there are many fields of application, HGR models have not reached their full potential, nor have they been widely adopted. This is caused mainly by three factors. First, the performance of HGR systems can still be improved (i.e., recognition accuracy and processing time, and number of gestures). Second, the protocol used for evaluating these models usually is poorly rigorous or ambiguous, and thus, the results are hardly comparable. Third, HGR implementations are commonly cumbersome. This is partly because they are not easy or intuitive to use (i.e., an HGR implementation is expected to be real-time, non-invasive, and wireless), or because they require some training or strict procedure before usage.

In this work, an HGR model focused on this third issue (procedures before usage, intuitive interface, and training/testing requirements) for HGR based on electromyography (EMG) signals is presented. In the following paragraphs, the problem is fully described.

### 1.1. Structure of Hand Gesture Recognition Systems

An HGR system is composed of five modules: data acquisition, pre-processing, feature extraction, classification, and post-processing. Data acquisition consists of measuring, via some physical sensors, the signals generated when a person performs a gesture [1]. All sorts of technologies have been used for data acquisition, such as inertial measurement units (IMUs) [19,20], cameras [21], force and flexion sensors (acquired through sensory gloves) [6,22], and sensors of electrical muscle activity (EMG) [23]. EMG signals can be captured via needle electrodes inserted in the muscle (intramuscular EMG, iEMG) or using surface electrodes which are placed over the skin (surface EMG, sEMG). The iEMG is used especially for medical diagnosis and has greater accuracy because needles can be directed on specific muscles [24]. On the other hand, sEMG is considered to be non-invasive. In this work, a non-invasive commercial device (Myo bracelet), which captures EMG signals, was used for data acquisition. EMG signals stand out among all other technologies because of their potential for capturing the intention of movement on amputees [25]. Pre-processing is the second module of an HGR system, which is in charge of organizing and homogenizing all sorts of acquired signals (i.e., sensor fusion) to match the feature extraction module. Common techniques used at this stage include filtering for noise reduction [7], normalization [26], or segmentation [27]. The next module of an HGR system is feature extraction. Its goal is to extract distinctive and non-redundant information from the original signals [28]. Features are intended to share similar patterns between elements of the same class. Feature extraction can be carried out using automatic feature extractors such as convolutional neural networks (CNNs) or autoencoders [29,30,31,32,33,34,35]. Other features can be selected manually with an arbitrary selection of the feature extraction functions. These functions can be extracted from time, frequency, or time–frequency domains [36]. However, most real-time HGR models use time-domain features because the controller delay for their computation is smaller compared to others. We found that the mean absolute value (MAV) was the most used feature for HGR applications. Nevertheless, we observed that other time-related features can also be used, such as root mean square (RMS), waveform length (WL), variance (VAR), fourth-order auto-regressive coefficients (AR-Coeff), standard deviation (SD), variance (VAR), energy ratio (ER), slope sign changes (SSC), mean, median, integrated EMG (iEMG), sample entropy (SampEn), mean absolute value ratio (MAVR), modified mean absolute value (MMAV), simple square integral (SSI), log detector (LOG), average amplitude change (AAC), maximum fractal length (MFL), dynamic time warping (DTW), sample entropy (SE), and quantization-based position weight matrix (QuPWM) [1,3,6,8,9,11,12,13,17,18].

The classifier module is composed of a supervised learning algorithm that maps a feature vector to a label. Common classifiers used for HGR applications are *k*-nearest neighbor [10], tree-based classifier [12], support vector machines (SVM) [6,11,37,38,39,40], Bayesian methods [41], neural networks (NN) [42,43,44], and recurrent neural networks [45,46,47,48]. Among these methods, it has been observed that SVM and CNN stand out, where SVM shows high efficiency with light computational requirements and fast responses, whereas CNN has very high recognition performance but requires hardware with more processing capacity and longer inference times. The last module is post-processing. Its objectives is to filter spurious predictions to produce a smoother response [49] and to adapt the responses of the classifier to final applications (e.g., a drone or robot).

### 1.2. Evaluation of Hand Gesture Recognition Systems

The performance of a hand gesture recognition system is analyzed based on three parameters: classification accuracy, recognition accuracy, and processing time. Classification and recognition concepts are differentiated in this work. Classification identifies the corresponding class of a given sample. The evaluation of classification just compares the predicted label with the true label of the EMG sample. Results of classification are usually presented in confusion matrices where sensitivity, precision, and accuracy are summarized by the gesture. Recognition goes further than classification because it not only involves assigning a sample to a label but also requires determining the instants of time where the gesture was performed. The evaluation of recognition accuracy, hence, compares the vector of predictions of an HGR system with the ground truth corresponding to the given EMG sample. The ground truth is a Boolean vector set over the points with muscle activity; this information is included in every sample of the data set, and it was obtained before by a manual segmentation procedure. There could be several ways of comparing the vector of predictions with the ground truth. In this work, the evaluation protocol previously defined in [50] is followed. This protocol calculates an overlapping factor between both vectors and considers a sample correctly recognized when the overlapping factor is above a threshold of 25%. This comparison is only carried out for a valid vector of predictions. A vector of predictions is valid when there is only one segment of continuous predictions with the same label which are different from the relax position. This can be considered as a strict evaluation because any point of the signal differently labeled will cause an incorrect recognition. Moreover, any relax label predicted in the middle of predictions of a different class will also imply an incorrect recognition. This way of evaluating recognition provides us with a true perspective of the HGR behavior in real applications. As a result, classification accuracy will be higher than recognition accuracy.

A demanding requirement for the HGR system is having real-time operation. For human–machine interfaces, a system works in real time when a person uses a system and does not perceive delay on the response [1]. This involves that real-time operation is dependent upon the application and user perception. There is much debate in the literature about the maximum time limit for a system to be considered in real time (e.g., 300 ms [51]). In this work, the threshold of 100 ms reported by [52] is considered. This time (also known as controller delay) is measured from the moment when the system receives the signal until it returns a response. Additionally, real-time operation is assured based on the time responses obtained over offline simulations. An offline simulation in this context is a simulation with previously obtained data. In contrast, an online evaluation involves new recordings of data every time it is going to be implemented. Additionally, HGR systems evaluated in online scenarios usually suffer from being tested over a small set of users (e.g., [53]). An offline evaluation has the advantage of using already collected data, and it also allows the experiments to be replicated and compared. An offline approach is suitable in our case where a large amount of data is required to evaluate the models. In our experiments, real-time data acquisition is simulated using a sliding window approach.

### 1.3. User-Specific and User-General HGR Systems

HGR systems are divided into two types: user-specific (dependent or individual models) and user-general (independent models). A user-specific system requires collecting samples each time a new user uses the system for training or tuning. On the other hand, user-general models are trained once over a multi-user data set, and these systems do not require additional collection of samples when a new user wants to use the system [54]. Although user-specific models are trained with fewer training samples, they usually obtain higher accuracies because they are trained and calibrated for each person. Meanwhile, user-general models are easier to use and set up. However, these models have a really low performance for a significant portion of users in the data set [29]. Developing user-general HGR systems is still an open research challenge because it requires not only large data sets but also robust and adaptable machine learning algorithms.

### 1.4. The Rotation Problem with Bracelet-Shaped Devices and Related Works

One of the main drawbacks of general HGR systems using a bracelet-shaped EMG devices is their dependence on the location of the sensor. This problem is usually diminished in the literature because HGR models are trained and evaluated assuming the exact location of the bracelet in the forearm of the user. In the literature, there are also reported examples of the downside effects of electrode displacement. For instance, Hargrove et al. [55] proposed a classifier training strategy in order to reduce the effect of electrode displacements on classification accuracy. Here, the system must be trained carefully. The samples corresponding to some rotation conditions were included in the training data. Sueaseenak et al. [56] proposed an optimal electrode position for the surface EMG sensor Myo bracelet. They found that the position to get the best surface EMG recording is in the middle of the forearm’s length area. This approach for wearing a bracelet sensor in its optimal position is not practical because it requires one to place the bracelet in exactly the same position every time the system is used. In [57], different experiments related to sensor orientation were applied when the testing data were shifted. The experiments demonstrated that shifting the sensor 2 cm causes the SVM’s and the kNN’s accuracy to drop significantly with accuracy between 50% and 60%. It is noticeable that sensor rotation decrements the performance of HGR systems and sometimes even makes those unusable. Therefore, it is important to have a system that corrects the variation in the orientation of the sensor. In this context, several researchers have tried to solve this problem with different methods. In [58], the bracelet was rotated every 45 degrees and the EMG signals were recorded. Then, a remapping was made according to the predicted angle and the distribution was marked on the user’s arm prior to the signal recording. However, the calculation time was high and it only worked well in steps of 45 degrees because of the high complexity of the algorithm. In [59], a classification system that uses the Myo bracelet and a correction to the rotation of the bracelet was applied showing a classification accuracy of 94.7%. However, the classification time was 338 ms, not applicable in real-time scenarios. Despite the fact that most of the previous works solve the problem of the sensor’s rotation found in the literature, the recognition was not evaluated in most of them, and only classification was performed. As a result, it is important to build a system that performs classification and recognition in conjunction with orientation correction.

### 1.5. Article Overview

The main contribution of this paper is the method for electrode rotation compensation, based on identifying the maximum energy channel (MEC) to detect the reference pod to compensate the variation in the orientation of the bracelet. The maximum energy is calculated using a reference hand gesture; then, the data are rearranged creating a new sensor order. This method is executed each time a person uses the recognition system, needing a maximum time of 4 s for the calibration process. After the calibration procedure, a person can use the proposed HGR system wearing the bracelet with a different rotation (i.e., any angle on the forearm). The proposed orientation correction algorithm was evaluated over a larger dataset following a stricter evaluation procedure for classification and recognition [50]. The data set has 612 users and was divided into two groups: 50% (i.e., 306 users) for training and 50% for testing. This work also implemented and compared user-specific and user-general models. One of the advantages of the HGR implemented system is its low computational cost and astonishing recognition and classification accuracy.

Following this introduction, the remaining of this paper is organized as follows. Section 2 presents Materials and Methods, including the EMG device used for collecting the data set, the gestures included, and the proposed model architecture to fix the displacement problem. In Section 3, the experiments designed for testing the proposed model are described. These include a comprehensive combination of user-specific and user-general models, original pod position and synthetic rotation, and HGR system with and without orientation correction. The results of these experiments are presented and analyzed in Section 4. In Section 5, further discussion over the results is presented. In Section 6, the findings of this research, as well as the outlines of the future work, are mentioned.

## 2. Materials and Methods

The architecture for the HGR system based on EMG signals that we developed in this work is presented in Figure 1. As can be observed, the proposed system is composed of five stages, which are data acquisition, pre-processing, feature extraction, classification, and post-processing. The mentioned stages are explained as follows.

### 2.1. Data Acquisition

This work uses the dataset collected in a previous research [60], and can be found in [61]. Additionally, the code has been uploaded to GitHub [62]. To simulate rotations of the bracelet, we assume that, by default, the pods of the Myo armband are ordered according to the sequence S=1,2,…,8. Then, with uniform probability, we randomly selected a number *r* from the set {−3,−2,−1,0,+1,+2,+3,+4}. Then, we simulated the rotation of bracelet by computing the new sequence $\tilde{S}=\tilde{s_{1}},\tilde{s_{2}},\ldots ,\tilde{s_{8}}$ of the pods, where $\tilde{s_{i}}=mod\left(s_{i}+r,9\right)$, with $s_{i}\in S$ and i=1,2,…,8. Note that in this way, we simulated rotations of the bracelet clockwise and counterclockwise in steps of 45 degrees.

The EMG signals were acquired with the Myo bracelet, which has eight differential electrodes with a sampling frequency of 200 Hz. This device also has an inertial measurement unit with nine degrees of freedom (accelerometer, gyroscope, and magnetometer) and haptic feedback, but in this work, we only used EMG information. The Myo bracelet is able to transmit the collected data via Bluetooth to a computer. The Myo bracelet sensor is illustrated in Figure 2a, the suggested manufacturer position of the Myo bracelet is observed in Figure 2b, and a sample of the Myo bracelet rotated in a different angle can be visualized in Figure 2c.

The protocol followed for acquiring EMG signals indicates that the Myo bracelet must be placed in the same area of the right or left forearm during the acquisition over all the users. In this research, the signals used are from people who wear the bracelet placed only on the right forearm, no matter if they were right- or left-handed. The data set is composed of 612 users and was divided into two groups: 50% for training and 50% for testing (i.e., 306 users for each one). It has to be noted that the data set is composed of 96% right-handed people and 4% left-handed people, as well as 66% men and 34% women. The age distribution of the data set has a higher concentration of users between 18 and 25 years old; this is because the data are from undergraduate students. An illustration of the statistical information related to the data set is presented in Figure 3.

The data set used in this work consists of five gestures, which are the same as those detected by the MYO manufacturer’s software. The mentioned hand gestures are waveIn, waveOut, fist, open, pinch, and the relax state (noGesture) as can be observed in Figure 4. The total number of repetitions performed by each user is 300, which corresponds to 50 repetitions for each gesture. Each repetition was recorded during 5 s, and every gesture repetition starts in the relax position and ends in the same relax position.

The data set also includes information on the limits of muscle activity, which was manually segmented within the 5 s of the measured EMG signal. This information is useful to identify the moments when every gesture was performed. For the rest of the paper, we use the name ground
Truth for the manual segmentation of the muscular activity.

#### 2.1.1. General and Specific Models

In this work, we train and evaluate two different approaches for hand gesture recognition based on a general and a specific model, respectively. We first created a general model based on a training set composed of EMG information from all users, and then each user tested the model to evaluate the recognition results. On the other hand, we also created a specific model based on a training set that only uses one user at a time, and again each user tested their respective model to evaluate the recognition results. To work with general or specific models, it is necessary to create a matrix organized per sensor, user, and gesture category to train the classifier. Equation (Equation 1) shows the EMG training matrix ${\mathrm{Dtrain}}_{user_{k}}$ for each user *k*.
(1)$${\mathrm{Dtrain}}_{user_{k}}=\left[\begin{array}{c} \mathrm{EMG}(user_{k},wog)\;\\ \;\mathrm{EMG}(user_{k},wig)\;\;\\ \mathrm{EMG}(user_{k},fg)\;\\ \;\mathrm{EMG}(user_{k},og)\\ \mathrm{EMG}(user_{k},pg)\;\\ \;\mathrm{EMG}(user_{k},ng) \end{array}\right]$$
where $\mathrm{EMG}(user_{k},gesture_{j})$ represents the EMG measures for each $user_{k}$ and $gesture_{j}$, waveOut (wog), waveIn (wig), fist (fg), open (og), pinch (pg), and noGesture (ng). Each matrix $\mathrm{EMG}(user_{k},gesture_{j})$ is composed of a set of the EMG measures denoted by ${\mathrm{Ms}}_{k}$, which represents the transposed vector of every channel repetition performed for $user_{k}$ as we show.
(2)$$\mathrm{EMG}\left(user_{k},gesture_{j}\right)=\left[{\mathrm{Ms}}_{1}\;{\mathrm{Ms}}_{2}\;\ldots \;{\mathrm{Ms}}_{8}\right]$$

Notice that the dimensions of each matrix are ${\mathrm{Ms}}_{k}\in R^{[P\times 7\times 6]\times 200}$, where we consider *P* as the number of the repetitions of a $gesture_{j}$, with seven sliding windows for each measure, six classes, and 200 extracted points for each sliding window that extract information of the EMG signal. It is worth mentioning that each sliding window was separated from each other by 25 points. Since the Myo sensor has eight EMG channels, we can write the EMG training matrix dimension as ${\mathrm{Dtrain}}_{{user}_{k}}\in R^{[[P\times 7\times 6]\times 200]\times 8}$.

Finally, the data of each user are appended in a general training matrix ${\mathrm{Dtrain}}_{total}$. When a user-general model is used, we consider (P=50). Equation (Equation 3) shows how a total training matrix for the user-general model is composed (k=306 users).
(3)$${\mathrm{Dtrain}}_{total}=\left[\begin{array}{c} {\mathrm{Dtrain}}_{{user}_{1}}\\ {\mathrm{Dtrain}}_{{user}_{2}}\\ \vdots \\ {\mathrm{Dtrain}}_{{user}_{k}} \end{array}\right]$$
where the EMG total training matrix dimension is ${\mathrm{Dtrain}}_{total}\in R^{[[[P\times 7\times 6]\times 200]\times Q]\times 8}$. The parameter *Q* represents the number of users used in the model. For the user-general model, Q=306, and for the user-specific model, Q=1. For the case of a user-specific model, the training matrix is composed only of signals belonging to each specific user. In user-specific models, the number of repetitions considered is P=25. It has to be noted that for each measure related to a $\mathrm{EMG}(user_{k}$ and $gesture_{j})$, a label Y $\in \left\{waveOut,waveIn,fist,open,pinch,andnoGesture\right\}$ is added to train the mode. *Y* denotes the label corresponding to the current EMG gesture sample, and to the seven sliding windows within it.

#### 2.1.2. Orientation Considerations for the EMG Sensor

In this research, two approaches were tested regarding the orientation problem of the Myo armband sensor, which are with and without orientation correction. Both methods were applied over the user-specific and user-general models previously explained.

Typically, the models—user-general and user-specific—that do not consider orientation correction present poor performance when the user places the bracelet in a different orientation. In this work, we propose an orientation correction algorithm to solve the problem related to the orientation variation of the Myo bracelet. This approach uses the maximum energy channel (MEC) of a $EMG_{gesture}$, which allows us to obtain high robustness to rotation and allows us to place the bracelet in any angle, similar to [63]. Furthermore, it helps to avoid the necessity to record the signals every time the system is going to be used.

For this purpose, a gesture to synchronize the HGR models was used. The synchronization gesture lets the sensor be used in a different position. All five gestures were tested as synchronization signals (sync). The results of the test for the selection of the best gesture for the synchronization signal are presented in Appendix A. These results demonstrated that the best performance was obtained using the waveOut gesture; thus, we selected that gesture for our experiments.

Performing the gesture waveOut during a period of time, a pod $S_{x}$ is obtained, which shows the location of the maximum activity in the ${\mathrm{EMG}}_{waveOut}$ signal. The EMG data are then rearranged according to $S_{x}$, obtaining a new sensor orientation for the HGR system. For this purpose, the average energy in every EMG window of 200 points is calculated for *T* repetitions, and then the maximum value is found in a specific pod. It is worth mentioning that one, two, three, or four windows of 200 points can be used as sync signals to identify $S_{x}$. The procedure to get the pod information in the synchronization stage starts with the data acquisition of the EMG signals of the sensor in the vector ${\mathrm{EMG}}_{wO}$, as we state as follows:(4)$${\mathrm{EMG}}_{wO}=\left[\begin{array}{cccccccc} s_{1} & s_{2} & s_{3} & s_{4} & s_{5} & s_{6} & s_{7} & s_{8} \end{array}\right]$$
where ${\mathrm{EMG}}_{wO}$$\in R^{200\times 8}$ and $s_{i}\in {\left[-1,1\right]}^{200\times 1}$. It has to be noted that the sample values from each channel $s_{i}$ are normalized values in the range of −1 and 1. Then, the energy of the samples of each channel is given by
(5)$$E_{wO}=\left[\begin{array}{cccccccc} E_{s_{1}} & E_{s_{2}} & E_{s_{3}} & E_{s_{4}} & E_{s_{5}} & E_{s_{6}} & E_{s_{7}} & E_{s_{8}} \end{array}\right]$$
where $E_{s}$ refers to the energy in each pod. The average energy $\bar{E_{s_{k}}}$ value over a channel for *T* repetitions of the gesture waveOut is represented by
(6)$$\bar{E_{s_{k}}}=\frac{1}{T}\sum_{j=1}^{T}\left(\sum_{i=2}^{L}abs\{\left(x_{i}\right)\cdot abs\left(x_{i}\right)-\left(x_{i-1}\right)\cdot abs\left(x_{i-1}\right)\}\right)$$
where abs refers to the absolute value, T∈[1,4] is the number of waveOut synchronization repetitions, k∈[1,8] represent the pod number, *L* is the length of the ${\mathrm{EMG}}_{waveOut}$ signal, and $x_{i}$ is the ith point of the ${\mathrm{EMG}}_{waveOut}$ signal. Then, the sensor $S_{x}$ is identified through the max function, which gives the maximum average energy value of the vector as we state as follows:(7)$$s_{x}=\max\left(\begin{array}{cccccccc} \bar{E_{s_{1}}} & \bar{E_{s_{2}}} & \bar{E_{s_{3}}} & \bar{E_{s_{4}}} & \bar{E_{s_{5}}} & \bar{E_{s_{6}}} & \bar{E_{s_{7}}} & \bar{E_{s_{8}}} \end{array}\right)$$

Finally, the new matrix order for all gestures is organized and described according the following equation:(8)$${\mathrm{EMG}}_{newOrder}=\left[\begin{array}{cccccc} s_{x} & s_{mod((x+1),9)} & s_{mod((x+2),9)} & \cdots & s_{mod((x+7),9)} & s_{mod((x+8),9)} \end{array}\right]$$
where mod refers to the remainder after division value, and the maximum value of (x+8) is 8 because there are eight pods. Notice that the default order coming from the Myo bracelet is as follows:(9)$${\mathrm{EMG}}_{defaultOrder}=\left[\begin{array}{cccccccc} s_{1} & s_{2} & s_{3} & s_{4} & s_{5} & s_{6} & s_{7} & s_{8} \end{array}\right]$$

As an example, if the $S_{x}$ detected is $S_{6}$, the new matrix is arranged as follows: ${\mathrm{EMG}}_{newOrder}=\left[\begin{array}{cccccccc} s_{6} & s_{7} & s_{8} & s_{1} & s_{2} & s_{3} & s_{4} & s_{5} \end{array}\right]$.

After obtaining the $S_{x}$ reference sensors through the maximum energy channel (MEC), we use it in training and testing procedures. It is important to highlight that the reference pod could not be the same for all recordings between users and gestures. The calibration process must be executed every time that a user wants to test the recognition system after the user takes the bracelet off.

For reproducing the results of the proposed models, the code and the dataset used for this paper are located in [64].

### 2.2. Pre-Processing

As part of the pre-processing stage, the EMG energy (Equation (Equation 15)) is used to identify if a current analyzed window needs to be classified or not. Every EMG window must exceed an energy threshold to be computed for the classifier. A threshold of 17% was considered in this research based on multiple tests with different energy thresholds. Whenever the energy of an analyzed window exceeds the threshold, the EMG window goes to the next stage, which is feature extraction. This process avoids the classification of unnecessary gestures if the threshold is not reached and, therefore, improves the computational cost. It has to be noted that the energy threshold is calculated using the synchronization gesture waveOut and adding consecutively the energy calculated from each channel to obtain the value of energy *E*.

To perform the pre-processing procedure, the eight pods of the Myo bracelet have been divided into two groups. Every group is composed of four pods—$group_{high}$ and $group_{low}$—that are analyzed individually with respect to the energy *E* and a threshold of 17%. The waveOut gesture requests a muscle activation pattern that is detected through the $group_{high}$. When a different gesture is performed, for example, waveIn, the activity is sensed through the $group_{low}$ of sensors. The channel division by groups allows the detection of gestures that activate a different group of muscles. The energy for $group_{high}$ corresponds to the energy of the pods $S_{1}$, $S_{2}$, $S_{3}$, $S_{4}$ as stated in Equation (Equation 10), while the energy for $group_{low}$ corresponds to the energy of the pods $S_{5}$, $S_{6}$, $S_{7}$, and $S_{8}$, as is shown in Equation (Equation 11).
(10)$$Th_{high}=\left(0.17\right)\frac{1}{4}\sum_{i=1}^{4}\bar{E_{S_{i}}}$$
(11)$$Th_{low}=\left(0.17\right)\frac{1}{4}\sum_{i=5}^{8}\bar{E_{S_{i}}}$$

### 2.3. Feature Extraction

Five functions to extract features are used in this paper, which are applied over every EMG recording (see Figure 5) contained into a sliding window only when it surpassed the threshold of energy.

The following set of functions that were used is briefly explained as follows:Standard deviation (SD): This feature measures the dispersion of the EMG signal. It indicates how the data are scattered respectively to the average and is expressed as:
(12)$$SD=\sqrt{\frac{1}{L-1}\sum_{i=1}^{L}|x_{i}-u{|}^{2}}$$
where $x_{i}$ is a sample of EMG signal, *u* is the average, and *L* is the total points of the EMG; Absolute envelope (AE): It uses the Hilbert transform for calculating the instantaneous attributes of a time series, especially amplitude and frequency [65]:
(13)$$AE=|AE|=\sqrt{{f\left(t\right)}^{2}+{\left(H\left\{f\left(t\right)\right\}\right)}^{2}}$$
where H(t) is the Hilbert transform and f(t) is the EMG signal; Mean absolute value (MAV): It is a popular feature used in EMG-based hand gesture recognition applications. The mean absolute value is the average of the absolute value of the EMG signal amplitude, and it is defined as follows:
(14)$$MAV=\frac{1}{L}\sum_{i=1}^{L}x_{i}$$
where $x_{i}$ is a sample of EMG signal, and and *L* is the total points of the EMG; Energy (E): It is a feature for measuring energy distribution, and it can be represented as [66]:
(15)$$E=\sum_{i=2}^{L}abs\{\left(x_{i}\right)\cdot abs\left(x_{i}\right)-\left(x_{i-1}\right)\cdot abs\left(x_{i-1}\right)\}$$
where $x_{i}$ is a sample of EMG signal, and *L* is total length of the EMG signal; Root mean square (RMS): It describes the muscle force and non-fatigue contraction [51]. Mathematically, the RMS can be defined as:
(16)$$RMS=\sqrt{\frac{1}{L}\sum_{i=1}^{L}{\left(x_{i}\right)}^{2}}$$
where $x_{i}$ is a sample of EMG signal, and *L* is the total points of the EMG.

### 2.4. Classification

A support vector machine (SVM) was chosen for the hand gesture classification. The SVM is a machine learning technique used to find the optimal separation hyper-plane in data classification [38,39,67]. It uses a kernel function in the input data to remap it into a new hyper-plane that facilitates the separation between classes. In this research, a polynomial kernel of third order with a one-vs.-one strategy was implemented to carried out the classification procedure. The parameters used to configure the SVM can be observed in Table 1. The parameters for the SVM were implemented in MATLAB for all the experiments.

For this research, the SVM multi-class classification was utilized. The multi-class problem is broken down to multiple binary classification cases, which is also called one-vs.-one coding [67]. The number of classifiers necessary for one-vs.-one multi-class classification can be retrieved with the formula n(n−1)/2, where *n* is the number of gesture classes.

In the one-vs.-one approach, each classifier separates points of two different classes, and uniting all one-vs.-one classifiers leads to a multi-class classifier. We use SVM since it is a classifier that allows portability of HGR systems due to its low computational cost and real-time operation [38,39,67]. In addition, in experiments conducted in [68,69], the authors demonstrate that SVM is able to reach a higher performance than *k*-nearest neighbor (KNN) for EMG signal classification.

In our research, the SVM training process was performed offline, obtaining different sets of support vectors for both user-specific and user-general models. It is worth mentioning that when we use a user-general model, the set of created support vectors influences the classifier inference time because this type of models were trained with a large amount of data. Therefore, more support vectors have to be analyzed before the classifier gives a response. When the SVM classifies an EMG window, a score matrix with values related to each gesture is generated, as is stated as follows:$$\mathrm{Scores}=\left[\begin{array}{cccccc} S_{g1}, & S_{g2}, & S_{g3}, & S_{g4}, & S_{g5}, & S_{g6} \end{array}\right]$$
where $S_{gi}$ is the corresponding score gesture value of $\left\{waveOut,waveIn,fist,open,pinch,noGesture\right\}$. The scores matrix is composed of negative scores, and the SVM gives as the selected label the one nearest to zero. These scores were turned into a positive range, as we can observe in the following equation,
$${\mathrm{Scores}}_{abs}=abs\left(\mathrm{Scores}\right),$$
and they are used to determine a maximum positive value each time a window is analyzed, as is presented as follows,
$$Scores_{max}=max\left({\mathrm{Scores}}_{abs}\right)$$
$${\mathrm{Scores}}_{norm}=\frac{{\mathrm{Scores}}_{abs}}{Scores_{max}}$$
(17)$$P_{s}=max\left(1-{\mathrm{Scores}}_{norm}\right)$$

Whenever a positive score $\left(P_{s}\right)$ value exceeds a threshold of 0.9 (based on different experiments), the label predicted by the classifier will be valid; otherwise, the default label is noGesture. Algorithm 1 for the operation of the SVM and the handling of the values of the scores matrix for each of the classification windows is as follows.
**Algorithm 1:** SVM Classification and Scores validation.
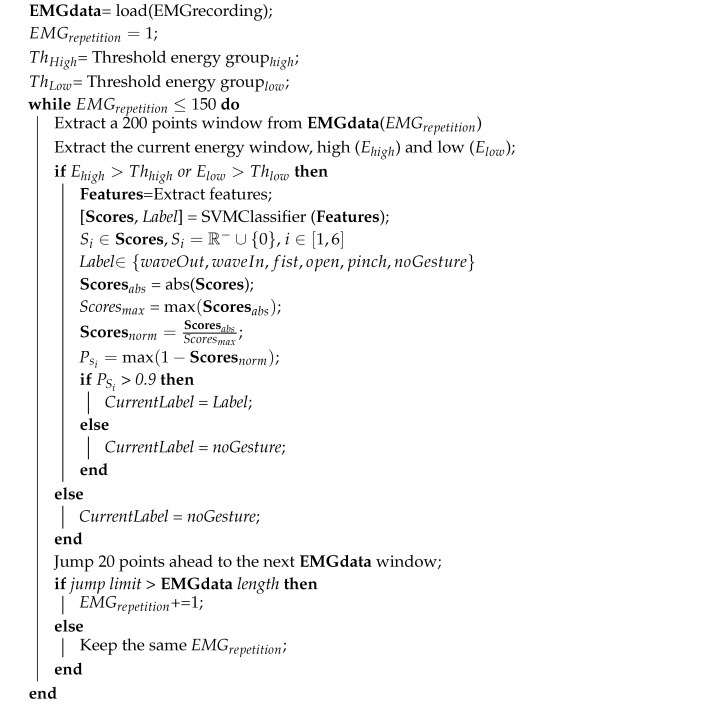


### 2.5. Post-Processing

During classification, each sliding window of 200 points with 20 points of separation was used to analyze the EMG signal, and then a vector with the probability of each gesture class was obtained, and only the most probable class was considered as the result of the classification stage. Then, the post-processing receives each of those class results, and a vector of labels is created by concatenating them. The vector of labels is finished when the number of sliding windows analyzed reaches the 5 s of recording. Then, we analyze the mode of every four labels, and the result is stored in a new vector of labels $B^{*}$, which is key to remove spurious labels that might appear during the classification results. In addition, we assign each those label results to a point in the time domain depending on the position of each sliding window. A sample of the vector of labels $B^{*}$ in the time domain is illustrated in Figure 6, where we can observe a set of noGesture labels, followed by a set of fist gesture labels, and again a set of noGesture labels. The ground truth $A^{*}$ can also be observed, which was obtained from the manual segmentation of the muscular activity that corresponds to a gesture. Finally, a recognition is considered successful if the vector of labels corresponds to the ground truth label, and if the vector of labels is aligned in time domain with the manual segmentation as illustrated in Figure 6. For this purpose, we used a minimum overlapping factor of ρ=0.25 as a threshold to decide if the recognition is correct. The overlapping factor is described in Equation (Equation 18),
(18)$$\rho =2\frac{|A^{*}\cap B^{*}|}{|A^{*}|+|B^{*}|}$$
where $A^{*}$ is the set of points where the muscle activity is located by the manual segmentation, and $B^{*}$ is the set of points where the gesture was detected by the model during post-processing.

## 3. Experimental Setup

The HGR classification and recognition experiments were carried out considering both user-specific and user-general models, and for each of them, we consider if each of those systems works with or without orientation correction. The information related to the experiments’ setup is illustrated in Figure 7. In addition, a brief explanation of each experiment can be found as follows.

Experiment 1: This experiment represents the ideal scenario suggested by the Myo bracelet manufacturer where each user trains and tests the recognition model, placing the bracelet in the same orientation recommended by the manufacturer. This orientation implies that a user should wear the bracelet in such a way that pod number 4 is always parallel to the palm of the hand (see Figure 2b). There is no orientation correction for this experiment;Experiment 2: The training EMG signals were acquired with the sensor placed in the orientation recommended by the manufacturer. However, when testing the model, the bracelet was rotated artificially (see Figure 2c). This experiment simulates the scenario where a user wears the sensor without taking into account the suggested positions for the testing procedure, which usually is the most common scenario. However, there is no orientation correction for this experiment;Experiment 3: The training EMG signals were acquired with the sensor placed in the orientation recommended by the manufacturer. For testing, the bracelet was rotated, simulating different angles. The orientation correction algorithm was applied for both training and testing data;Experiment 4: In this experiment, the performance of the proposed method is evaluated when there is rotation of the bracelet for training and testing, and the orientation correction algorithm was applied for both training and testing data.

## 4. Results

In this section, we present the HGR performance results for the Myo armband sensor manufacturer’s model, as well as our results for the user-specific and user-general models. In addition, we also compare our user-specific and user-general results with each other, and then we compare such results with other approaches that can be found in the literature. For this purpose, we use confusion matrices where accuracy, precision, and sensitivity information values can be visualized.

To calculate the accuracy, the number of true positives (TP) values are divided by the total set of samples analyzed, which includes true positives (TP), true negatives (TN), false positives (TP), and false negatives (TP). The accuracy value, which is considered our main metric of evaluation, is useful to analyze the proportion of correct predictions over a set of measures, as can be observed in Equation (Equation 19).
(19)$$Accuracy=\frac{TP}{TP+TN+FP+FN}\times 100\%$$

We also calculated the sensitivity and precision values as support metrics of evaluation. The sensitivity (also known as recall) is the fraction of the total amount of relevant instances that were actually retrieved—i.e., how many recognized gestures are relevant. On the other hand, the precision (also called positive predictive value) is the fraction of relevant instances among the retrieved instances—i.e., how many relevant gestures are recognized. The sensitivity and precision metrics can be observed in Equations (Equation 20) and (Equation 21), respectively.
(20)$$Sensitivity=\frac{TP}{TP+FN}\times 100\%$$
(21)$$Precision=\frac{TP}{TP+FP}\times 100\%$$

### 4.1. Myo Bracelet Model Results Using Manufacturer’s Software

The classification results obtained using the MYO bracelet manufacturer’s model are presented in Table 2. It is worth mentioning that the Myo bracelet manufacturer’s recognition system provides an answer every 20 ms. As can be observed, the accuracy obtained for classification is 64.66% using the suggested position by the manufacturer.

### 4.2. User-Specific HGR Model Result

The classification results for $experiment_{1}$, $experiment_{2}$, $experiment_{3}$, and $experiment_{4}$ for the user-specific models are presented in Table 3, Table 4, Table 5 and Table 6, respectively. As can be observed, the classification accuracy obtained was 94.99% for $experiment_{1}$, 39.38% for $experiment_{2}$, 94.93% for $experiment_{3}$, and 94.96% for $experiment_{4}$. The worst possible scenario was $experiment_{2}$ with a classification accuracy of 39.38%. This is because the bracelet sensor was rotated for the test set, and there was no orientation correction for this experiment. On the other hand, the best result among all the experiments for user-specific models was $experiment_{4}$ with a classification accuracy of 94.96%. This is usually the most common scenario that can be present during the experiments because it takes into account simulated rotation during training and testing. The approach used for $experiment_{4}$ also considered the orientation correction, which helps to achieve high classification results. The best precision and sensitivity results were obtained during $experiment_{4}$ for the waveIn gesture with 98.89% and the waveOut gesture with 97.66%, respectively. It has to be noted that we present only the best results for $experiment_{3}$ and $experiment_{4}$, which were obtained with four synchronization gestures (sync=4) to select the maximum average energy sensor $s_{x}$. The other results for (sync =1, 2, and 3) can be found in Appendix B.

### 4.3. User-General HGR Model Results

The classification results for $experiment_{1}$, $experiment_{2}$, $experiment_{3}$, and $experiment_{4}$ for the user-general models are presented in Table 7, Table 8, Table 9 and Table 10, respectively. As can be observed, the classification accuracy obtained was 81.6% for $experiment_{1}$, 44.52% for $experiment_{2}$, 81.2% for $experiment_{3}$, and 81.22% for $experiment_{4}$. The worst scenario was $experiment_{2}$ with a classification accuracy of 44.52%. This is because the bracelet sensor was rotated for the test set, and there was no orientation correction for this experiment. On the other hand, the best result among all the experiment for user-specific models was $experiment_{4}$ with a classification accuracy of 81.22%. This is usually the most common scenario that can be present during the experiments because it takes into account simulated rotation during training and testing. The approach used for $experiment_{4}$ also considered the orientation correction, which helps to achieve high classification results. The best precision and sensitivity results were obtained during $experiment_{4}$ for the pinch gesture with 88.02% and the noGesture gesture with 89.9%, respectively. It has to be noted that we present only the best results for $experiment_{3}$ and $experiment_{4}$, which were obtained with four synchronization gestures (sync=4) to select the maximum average energy sensor $s_{x}$. The other results for (sync = 1, 2, and 3) can be found in Appendix C.

### 4.4. Comparison between User-Specific and User-General Results

In this section, we summarize and compare the best classification results obtained from the HGR proposed system. We also include in this section the recognition results for each experiment, which are obtained after the post-processing stage. Both classification and recognition are presented in terms of accuracy. In Figure 8, we present the results for all the users without taking into account sex or handedness preference information. Figure 9 presents the results considering the user’s sex, and Figure 10 presents the results considering handedness preferences. The presented results correspond to the best for each experiment, which means that for $experiment_{1}$ and $experiment_{2}$, there is no synchronization gesture (sync=0), and for $experiment_{3}$ and $experiment_{4}$, we used four synchronization gestures (sync=4) to select the maximum average energy sensor $s_{x}$.

As can be seen in Figure 8, when the user-general model is used, the accuracy of the system without taking into account sex or handedness preference information decreases by up to 13.7% for classification and up to 13.9% for recognition, respectively. It also decreases by up to 15.9% for classification and 15.9% for recognition for the experiments considering the user’s sex—Figure 9. Moreover, its accuracy also decreases by up to 16.7% for classification and 16.9% for recognition for the experiments considering handedness preference—Figure 10. However, it is observed in Figure 8 that, in general, only for $experiment_{2}$, the user-general model obtains slightly better results than in the other experiments—up to 7.6% better. Nevertheless, $experiment_{2}$ also obtains the worst results for classification—from 39.8% to 44.5%—and recognition—from 38.8% to 43.4%. This behavior is repeated in Figure 9 and Figure 10. The observed decrease in accuracy when using a user-general model is a common behavior in classification and recognition systems. This is because typically, the performance tends to decrease when a large data set is used to analyze the generalization properties of a proposed model. For this reason, and based on the aforementioned results, we consider that the generalization capabilities of the proposed HGR system are acceptable since its performance does not decrease drastically when we compare user-specific with user-general models.

To analyze the effect of the orientation correction algorithm over all the experiments, we focus on the general data results presented in Figure 8. It can be seen that when the orientation correction is used, the performance is capable of increasing classification and recognition performances up to 45.4% and 36.9%, respectively. This indicates that the orientation correction approach has a positive and substantial impact on the performance of the HGR models. This behavior is repeated in Figure 9 and Figure 10.

In order to analyze the user’s sex-related results over all the experiments, we focus on the results presented in Figure 9. It can be observed that women obtain better results in the user-specific model—up to 1.6% better—while men obtain better results in the user-general model—up to 3.1% better. This might be due to the fact that there are more men—66%—than women—44%—in the overall data set, which decreases the performance of women when using the user-general models.

To analyze the user’s handedness preference-related results over all experiments, we focus on the results presented in Figure 10. It can be observed that left-handed users present better results in the user-specific model—up to 8.1% better—while right-handed users present better results for the user-general model—up to 3.6% better. This might be due to the fact that there are more right-handed users—96%—than left-handed users—4%—in the overall data set, which decreases the performance of left-handed users when using the user-general models.

Finally, in Table 11, we show the average classification time for the user-general and user-specific models. It can be observed that the average time in the user-general models is higher than in the user-specific case. This is because the general model is composed of several data users and there is a greater number of support vectors that must be analyzed before the classifier gives a label response. However, the response time of both the user-specific and user-general models is close to 100 ms, which is considered real-time for this application.

### 4.5. Comparison of Results with Other Papers

We compare our user-specific and user-general HGR models with other proposals in terms of classification and recognition in Table 12. Although recognition evaluation is mentioned in those proposals, in most of them, only classification was performed. Moreover, several experiments performed in these papers were carried out without sensor rotation considerations. For example, a rotation correction was performed in [63], but such work does not evaluate recognition. Another approach is presented in [59], where no recognition evaluation was presented, but a rotation correction algorithm was proposed.

As can be observed, our proposed user-general model obtained better results compared to [55,57,70]. Moreover, our user-general system performed better, even when training a model based on 306 users, while the others only trained their models as a user-specific approach. On the other hand, our user-specific model also obtained better results compared to [55,57,58,59,70], which are also user-specific-based models. The only approach that obtained better results than our proposed approach is [63]. However, that approach does not use a recognition criterion for evaluation, and it trained and tested the model using only 40 users. It does not help to compare its generalization capabilities with those of our proposed model, which uses 306 users for training and testing, respectively.

## 5. Discussion

During the experiments, we noticed that the recognition performance for most of the experiments is significantly lower than the classification performance. This is because for classification, the time in which a gesture is executed is not relevant. On the other hand, recognition requires information about the time when the gestures were detected. This is a key aspect since recognition needs to have a minimum overlap among the predicted and ground-truth signals to indicate that the prediction of a gesture was successful.

The best classification and recognition results were obtained during $experiment_{1}$ for both user-specific and user-general models—see Figure 8. During $experiment_{1}$, the users always wore the Myo bracelet in exactly the same orientation and following the considerations of the Myo manufacturer for both training and testing, which can be considered an ideal scenario. Thus, the accuracy results obtained during $experiment_{1}$ for classification are 95% and 81.6% for the user-specific and user-general models, respectively. On the other hand, the accuracy results for recognition are 94.2% and 80.6% for the user-specific and user-general models, respectively. Nevertheless, $experiment_{4}$ reached almost the same results using the orientation correction algorithm, even if the bracelet was rotated for the training and test sets, which demonstrated the effectiveness of the proposed orientation correction algorithm. The accuracy results obtained during $experiment_{4}$ for classification are 95% and 81.2% for the user-specific and user-general models, respectively. On the other hand, the accuracy results for recognition are 94.2% and 80.3% for the user-specific and user-general models, respectively.

The worst classification and recognition results were obtained during $experiment_{2}$ for both user-specific and user-general models. During $experiment_{2}$, the users changed the angles of the Myo bracelet for the testing procedure, and there was no orientation correction performed, which can be considered the worst possible scenario. The accuracy results obtained during $experiment_{2}$ for classification are 39.8% and 44.5% for the user-specific and user-general models, respectively. On the other hand, the accuracy results for recognition are 38.8% and 43.4% for the user-specific and user-general models, respectively.

For $experiment_{3}$, we started to notice the positive effects of using the orientation correction approach, which allowed us to increase the accuracy results for both user-specific and user-general models. During $experiment_{3}$, the sensor was not rotated for training, but it was rotated for testing, and the orientation correction was applied on both training and testing data. The accuracy results obtained during $experiment_{3}$ for classification are 94.9% and 81.2% for the user-specific and user-general models, respectively. On the other hand, the accuracy results for recognition are 94.2% and 80.3% for the user-specific and user-general models, respectively. Since the only difference between $experiment_{2}$ and $experiment_{3}$ was that the latter used orientation correction in training and testing data, $experiment_{3}$ was useful to evaluate the effect of orientation correction. If we compare with $experiment_{2}$, the performance of $experiment_{3}$ increased classification accuracy up to 55.1% and 36.7% for user-specific and user-general models, respectively. Moreover, similar behavior was presented for recognition performance, which increased up to 55.4% and 37% for user-specific and user-general models, respectively. This suggests that the orientation correction approach has a positive and substantial impact on the performance of the HGR models.

In $experiment_{4}$, we also observed the positive effects of using the orientation correction approach, which allowed us to increase the accuracy results for both user-specific and user-general models. During $experiment_{4}$, the sensor was rotated for both training and testing data, and the orientation correction was also applied on both of them. The accuracy results obtained during $experiment_{4}$ for classification are 95% and 81.2% for the user-specific and user-general models, respectively. On the other hand, the accuracy results for recognition are 94.2% and 80.3% for the user-specific and user-general models, respectively. These results suggest that although the Myo sensor was rotated for the train and test data for $experiment_{4}$, we obtained similar results comparable to $experiment_{3}$ where only the sensor was moved for the test. This suggests that the orientation correction approach has a positive and substantial impact on the performance of the HGR models even if the train and test sets were collected with the Myo sensor rotated.

The results obtained using the Myo sensor manufacturer’s model show an acceptable performance as long as the bracelet is placed in the suggested position. However, the proposed user-specific and user-general models considerably improved the performance of the Myo bracelet even when there was rotation of the Myo bracelet on the train and test sets. If we compare with the Myo sensor manufacturer’s model results, the performance of $experiment_{4}$ increases classification accuracy up to 30.6% and 16.6% for user-specific and user-general models, respectively. Moreover, a similar behavior is presented for recognition performance, which increased up to 29.5% and 15.7% for user-specific and user-general models, respectively.

Usually, the classification and recognition performance tends to decrease when a large data set is used to analyze the generalization properties of an HGR model. For this reason, we observed during the experiments that the performance of the HGR model decreases when using a user-general model. However, such a performance does not decrease drastically when using a user-general model; thus, we consider that the generalization capabilities of the proposed HGR system are acceptable.

During the experiments, it was observed that to obtain promising results, the correct selection of the synchronization gesture was a key point. Better results were obtained for $experiment_{3}$ and $experiment_{4}$ when the synchronization gesture was repeated four times (sync=4) for the correct selection of the maximum average energy sensor $s_{x}$.

## 6. Conclusions

In this work, a method to correct the orientation rotation of the Myo bracelet sensor for user-specific and user-general hand gesture recognition models was presented. The algorithm for the correction of orientation is based on finding the maximum energy channel for a set of synchronization EMG samples of the gesture waveOut. Based on the maximum average energy sensor $s_{x}$ predicted for the orientation correction algorithm, a new order of the sensor pods is obtained, and then the Myo bracelet sensor pods information are realigned accordingly to consider the sensor with more energy. Our experiments evaluated user-specific and user-general hand gesture recognition models combined with artificial rotations of the bracelet. The classification and recognition results obtained were encouraging. The proposed orientation correction algorithm can improve the classification and recognition performance of the hand gesture recognition system, even if the Myo bracelet sensor is rotated during the training and test sets.

Although the obtained results were promising, there are still improvements that can be made on the user-specific and user-general model performance that might allow us to finetune our method in future works—for example, testing more sophisticated classifiers, improving feature extraction, and using a different post-processing method, among others.

## Figures and Tables

**Figure 1 sensors-20-06327-f001:**
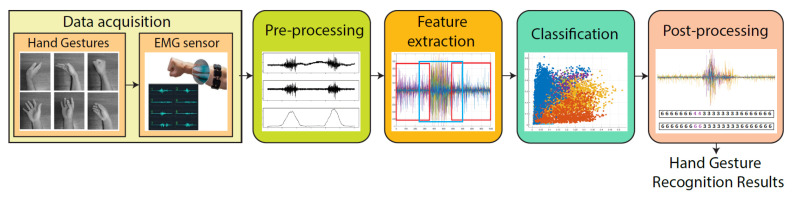
Hand gesture recognition architecture. It can be observed that the proposed architecture is composed of five stages, which are data acquisition, pre-processing, feature extraction, classification, and post-processing.

**Figure 2 sensors-20-06327-f002:**
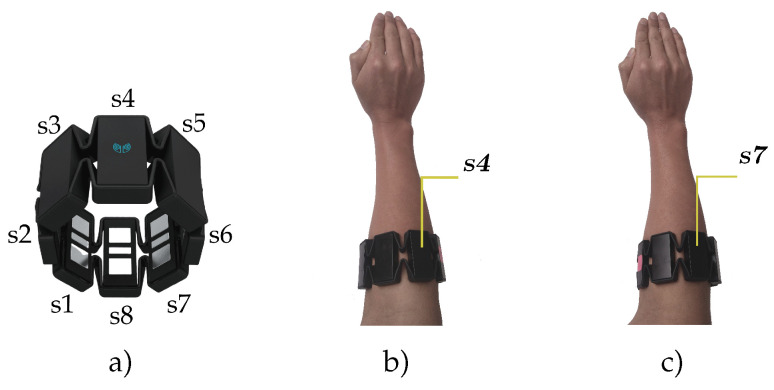
Myo armband sensor. (**a**) Myo pod distribution, (**b**) position of the sensor suggested by the Myo manufacturer, and (**c**) position of the Myo sensor rotated, which can cause issues during the recognition procedure.

**Figure 3 sensors-20-06327-f003:**
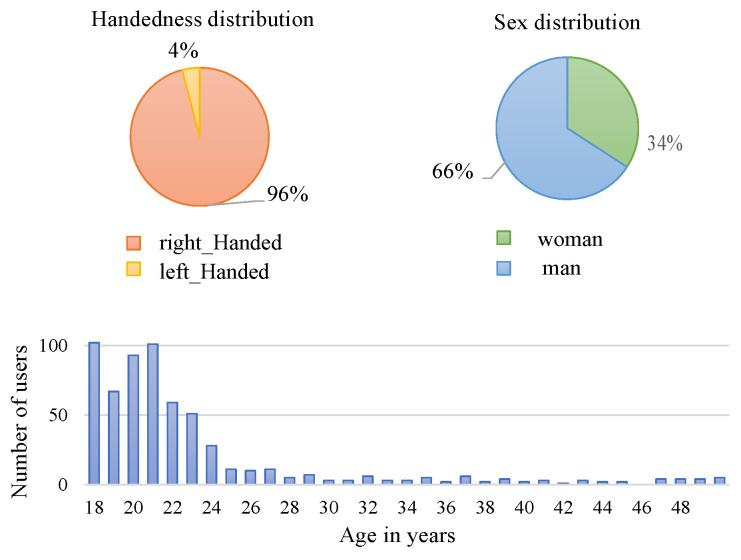
Data set statistics related to handedness distribution, sex, and age. The illustrations refer to the total number of users—612—in the data set. The data set is divided into 50% of users for training and 50% for test—306 for each, respectively.

**Figure 4 sensors-20-06327-f004:**
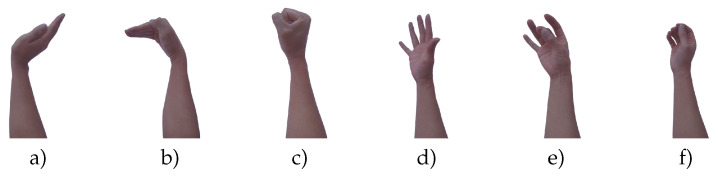
Hand gestures to be recognized for the proposed architecture. (**a**) waveOut, (**b**) waveIn, (**c**) fist, (**d**) open, (**e**) pinch, and (**f**) noGesture.

**Figure 5 sensors-20-06327-f005:**
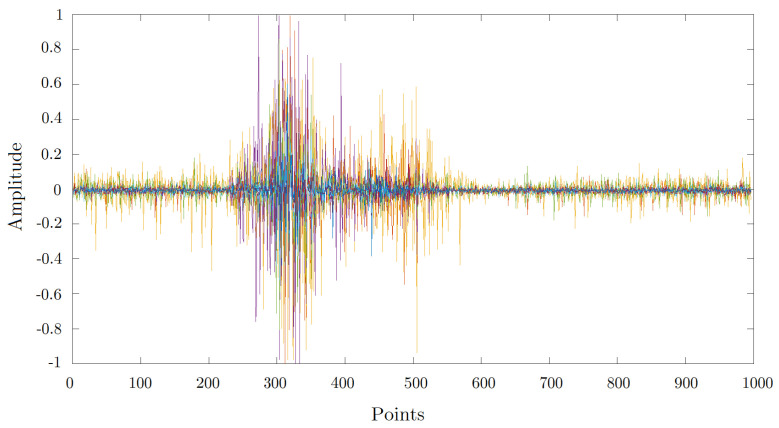
A sample of an electromyography (EMG) signal recorded using the Myo bracelet with the position of the sensors suggested by the Myo manufacturer for the Fist gesture.

**Figure 6 sensors-20-06327-f006:**
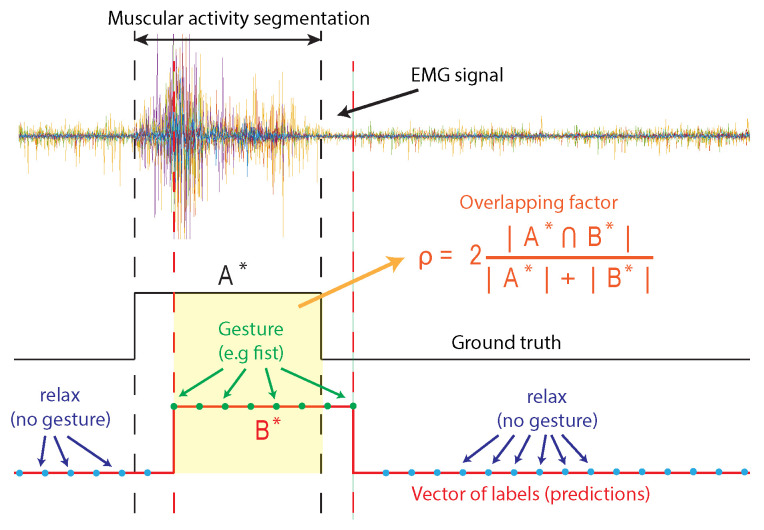
Calculation of value ρ through overlapping among ground-truth and the vector of predictions. If overlapping factor for each EMG sample is more than ρ=0.25, then we consider that the recognition is correct.

**Figure 7 sensors-20-06327-f007:**
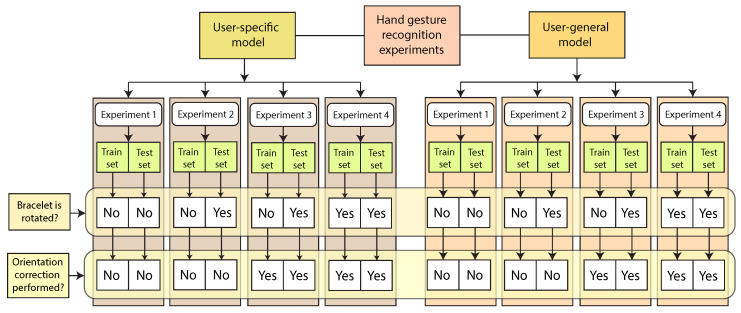
Experiment setup diagram. We performed our experiments using user-specific and user-general models, and for each one of them, we evaluated the bracelet rotation with and without the proposed orientation correction method.

**Figure 8 sensors-20-06327-f008:**
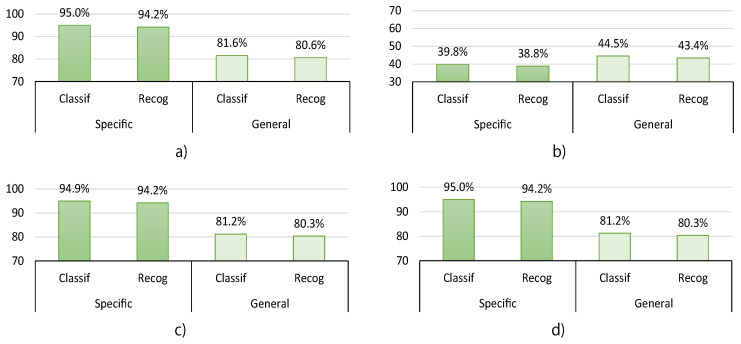
Hand gesture recognition (HGR) classification and recognition accuracy results for all users without taking into account sex or handedness preference information for user-specific and user-general models obtained for (**a**) $experiment_{1}$, (**b**) $experiment_{2}$, (**c**) $experiment_{3}$, and (**d**) $experiment_{4}$.

**Figure 9 sensors-20-06327-f009:**
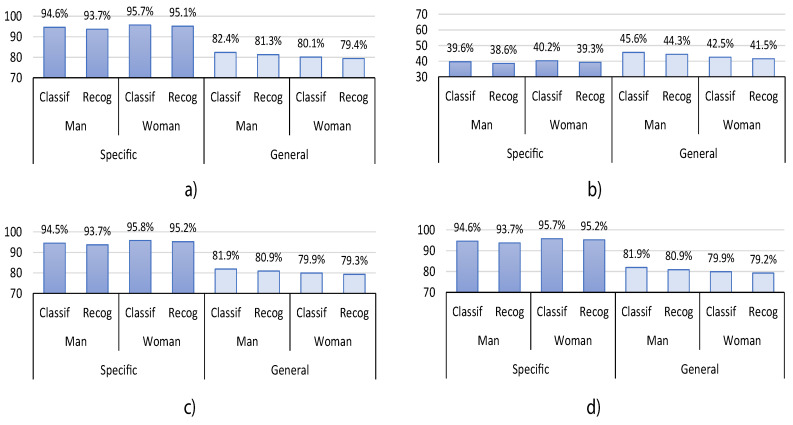
HGR classification and recognition accuracy results considering user’s sex information for user-specific and user-general models obtained for (**a**) $experiment_{1}$, (**b**) $experiment_{2}$, (**c**) $experiment_{3}$, and (**d**) $experiment_{4}$.

**Figure 10 sensors-20-06327-f010:**
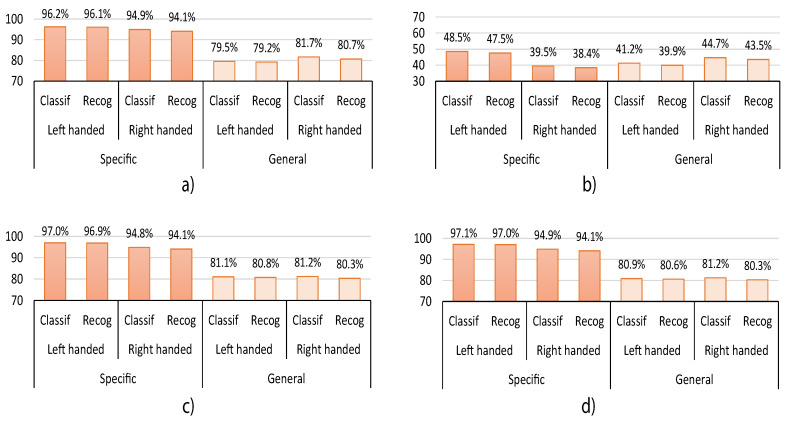
HGR classification and recognition accuracy results considering handedness preference for user-specific and user-general models obtained for (**a**) $experiment_{1}$, (**b**) $experiment_{2}$, (**c**) $experiment_{3}$, and (**d**) $experiment_{4}$.

**Table 1 sensors-20-06327-t001:** Support vector machine (SVM) configuration.

MATLAB Variable	Value
Kernel Function	polynomial
Polynomial Order	3
Box Constrain	1 (variable value for regularization)
Standardize	$\left({Feature}_{i}-\mu \right)/\sigma$; where μ = mean, σ = standard deviation
Coding	one vs one

**Table 2 sensors-20-06327-t002:** Confusion matrix of the Myo bracelet using the manufacturer’s model and suggested sensor position. Classification accuracy =64.66%.

	Targets						Predictions Count (Precision%)
	waveIn	waveOut	Fist	Open	Pinch	noGesture	
waveIn	**4831**	431	164	211	218	3	**5858** **82.47%**
waveOut	368	**5370**	262	682	406	3	**7091** **75.73%**
fist	1047	548	**5361**	1009	1588	29	**9582** **55.95%**
open	334	458	404	**4072**	795	2	**6065** **67.14%**
pinch	105	253	337	342	**2437**	3	**3477** **70.09%**
noGesture	965	590	1122	1334	2206	**7610**	**13827** **55.04%**
**Targets Count** **(Sensitivity%)**	**7650** **63.15%**	**7650** **70.2%**	**7650** **70.08%**	**7650** **53.23%**	**7650** **31.86%**	**7650** **99.48%**	**45,900** **64.66%**

**Table 3 sensors-20-06327-t003:** Confusion matrix of $experiment_{1}$ for the user-specific model. Rotation of the bracelet =NO, orientation correction =NO. The sync gesture was not used. Classification accuracy =94.99%.

	Targets						Predictions Count (Precision%)
	waveIn	waveOut	Fist	Open	Pinch	noGesture	
waveIn	**7339**	65	73	57	36	168	**7738** **94.84%**
waveOut	86	**7416**	64	54	32	136	**7788** **95.22%**
fist	18	10	**7305**	43	19	136	**7531** **97%**
open	79	94	100	**7385**	113	138	**7909** **93.37%**
pinch	34	41	53	49	**7232**	150	**7559** **95.67%**
noGesture	94	24	55	62	218	**6922**	**7375** **93.86%**
**Targets Count** **(Sensitivity%)**	**7650** **95.93%**	**7650** **96.94%**	**7650** **95.49%**	**7650** **96.54%**	**7650** **94.54%**	**7650** **90.48%**	**45,900** **94.99%**

**Table 4 sensors-20-06327-t004:** Confusion matrix of $experiment_{2}$ for the user-specific model. Rotation of the bracelet =YES (on the test set), orientation correction =NO. The sync gesture was not used. Classification accuracy =39.83%.

	Targets						Predictions Count (Precision%)
	waveIn	waveOut	Fist	Open	Pinch	noGesture	
waveIn	**2961**	2265	2104	2231	2155	291	**12007** **24.66%**
waveOut	1204	**2320**	970	1030	756	136	**6416** **36.16%**
fist	1763	1874	**2862**	1714	1579	254	**10046** **28.49%**
open	515	526	594	**1389**	516	127	**3667** **37.88%**
pinch	869	566	874	965	**2052**	143	**5469** **37.52%**
noGesture	338	99	246	321	592	**6699**	**8295** **80.76%**
**Targets Count** **(Sensitivity%)**	**7650** **38.71%**	**7650** **30.33%**	**7650** **37.41%**	**7650** **18.16%**	**7650** **26.82%**	**7650** **87.57%**	**45,900** **39.83%**

**Table 5 sensors-20-06327-t005:** Confusion matrix of $experiment_{3}$ for the user-specific model. Rotation of the bracelet =YES (on the test set), orientation correction =YES. Best result with four synchronization gestures (sync=4) to select the maximum average energy sensor $s_{x}$. Classification accuracy =94.93%.

	Targets						Predictions Count (Precision%)
	waveIn	waveOut	Fist	Open	Pinch	noGesture	
waveIn	**7338**	49	80	46	39	171	**7723** **95.01%**
waveOut	75	**7460**	65	54	29	134	**7817** **95.43%**
fist	22	13	**7301**	44	22	137	**7539** **96.84%**
open	76	68	118	**7381**	123	139	**7905** **93.37%**
pinch	31	40	43	36	**7175**	149	**7474** **96%**
noGesture	108	20	43	89	262	**6920**	**7442** **92.99%**
**Targets Count** **(Sensitivity%)**	**7650** **95.92%**	**7650** **97.52%**	**7650** **95.44%**	**7650** **96.48%**	**7650** **93.79%**	**7650** **90.46%**	**45,900** **94.93%**

**Table 6 sensors-20-06327-t006:** Confusion matrix of $experiment_{4}$ for the user-specific model. Rotation of the bracelet =YES (on training and test set), orientation correction =YES. Best result with four synchronization gestures (sync=4) to select the maximum average energy sensor $s_{x}$. Classification accuracy =94.96%.

	Targets						Predictions Count (Precision%)
	waveIn	waveOut	Fist	Open	Pinch	noGesture	
waveIn	**7335**	50	86	46	40	173	**7730** **94.89%**
waveOut	77	**7471**	59	50	28	134	**7819** **95.55%**
fist	27	10	**7307**	43	24	141	**7552** **96.76%**
open	72	67	113	**7386**	125	137	**7900** **93.49%**
pinch	33	35	41	33	**7174**	150	**7466** **96.09%**
noGesture	106	17	44	92	259	**6915**	**7433** **93.03%**
**Targets Count** **(Sensitivity%)**	**7650** **95.88%**	**7650** **97.66%**	**7650** **95.52%**	**7650** **96.55%**	**7650** **93.78%**	**7650** **90.39%**	**45,900** **94.96%**

**Table 7 sensors-20-06327-t007:** Confusion matrix of $experiment_{1}$ for the user-general model. Rotation of the bracelet =NO, orientation correction =NO. The sync gesture was not used. Classification accuracy =81.6%.

	Targets						Predictions Count (Precision%)
	waveIn	waveOut	Fist	Open	Pinch	noGesture	
waveIn	**6421**	151	201	239	549	186	**7747** **82.88%**
waveOut	198	**6544**	112	516	270	134	**7774** **84.18%**
fist	467	26	**6696**	278	682	153	**8302** **80.66%**
open	209	799	358	**6070**	891	170	**8497** **71.44%**
pinch	160	79	173	395	**4832**	116	**5755** **83.96%**
noGesture	195	51	110	152	426	**6891**	**7825** **88.06%**
**Targets Count** **(Sensitivity%)**	**7650** **83.93%**	**7650** **85.54%**	**7650** **87.53%**	**7650** **79.35%**	**7650** **63.16%**	**7650** **90.08%**	**45,900** **81.6%**

**Table 8 sensors-20-06327-t008:** Confusion matrix of $experiment_{2}$ for the user-general model. Rotation of the bracelet =YES (on the test set), orientation correction =NO. The sync gesture was not used. Classification accuracy =44.52%.

	Targets						Predictions Count (Precision%)
	waveIn	waveOut	Fist	Open	Pinch	noGesture	
waveIn	**3490**	3049	2426	2991	2986	413	**15355** **22.73%**
waveOut	1333	**3007**	561	488	189	100	**5678** **52.96%**
fist	1619	715	**3437**	1362	1775	203	**9111** **37.72%**
open	341	645	561	**2146**	710	91	**4494** **47.75%**
pinch	568	117	422	377	**1594**	81	**3159** **50.46%**
noGesture	299	117	243	286	396	**6762**	**8103** **83.45%**
**Targets Count** **(Sensitivity%)**	**7650** **45.62%**	**7650** **39.31%**	**7650** **44.93%**	**7650** **28.05%**	**7650** **20.84%**	**7650** **88.39%**	**45,900** **44.52%**

**Table 9 sensors-20-06327-t009:** Confusion matrix of $experiment_{3}$ for the user-general model. Rotation of the bracelet =YES (on the test set), orientation correction =YES. Best result with four synchronization gestures (sync=4) to select the maximum average energy sensor $s_{x}$. Classification accuracy =81.2%.

	Targets						Predictions Count (Precision%)
	waveIn	waveOut	Fist	Open	Pinch	noGesture	
waveIn	**6666**	143	344	296	744	215	**8408** **79.28%**
waveOut	197	**6482**	85	370	260	139	**7533** **86.05%**
fist	341	39	**6612**	251	663	160	**8066** **81.97%**
open	163	892	387	**6257**	1069	170	**8938** **70%**
pinch	92	30	121	265	**4373**	87	**4968** **88.02%**
noGesture	191	64	101	211	541	**6879**	**7987** **86.13%**
**Targets Count** **(Sensitivity%)**	**7650** **87.14%**	**7650** **84.73%**	**7650** **86.43%**	**7650** **81.79%**	**7650** **57.16%**	**7650** **89.92%**	**45,900** **81.2%**

**Table 10 sensors-20-06327-t010:** Confusion matrix of $experiment_{4}$ for the user-general model. Rotation of the bracelet =YES (on training and test set), orientation correction =YES. Best result with four synchronization gestures (sync=4) to select the maximum average energy sensor $s_{x}$. Classification accuracy =81.22%.

	Targets						Predictions Count (Precision%)
	waveIn	waveOut	Fist	Open	Pinch	noGesture	
waveIn	**6651**	138	336	302	725	217	**8369** **79.47%**
waveOut	207	**6550**	83	416	264	139	**7659** **85.52%**
fist	359	29	**6614**	262	656	160	**8080** **81.86%**
open	147	849	391	**6165**	1034	162	**8748** **70.47%**
pinch	95	30	126	295	**4424**	95	**5065** **87.34%**
noGesture	191	54	100	210	547	**6877**	**7979** **86.19%**
**Targets Count** **(Sensitivity%)**	**7650** **86.94%**	**7650** **85.62%**	**7650** **86.46%**	**7650** **80.59%**	**7650** **57.83%**	**7650** **89.9%**	**45,900** **81.22%**

**Table 11 sensors-20-06327-t011:** Average classification time.

Model	Specific	General
Time (ms)	16.97.2±17.52	71.69±54.76

**Table 12 sensors-20-06327-t012:** Classification and recognition comparisons.

Paper	Device	Pods Sensors	Gestures		Train/Test Users	Class.(%)	Recog.(%)	HGR Model	Recognition Evaluated	Rotation Performed	Correction of Rotation
[39]	MYO	8 *	5	$Gr_{1}$	12/12	97.80	-	S	no	no	no
[70]	Delsys	12	6	$Gr_{3}$	40/40	79.68	-	S	no	no	no
[39]	MYO	8 *	5	$Gr_{1}$	12/12	98.70	-	S	no	no	no
[55]	Sensors	5	11	$Gr_{4}$	4/4	81.00	-	S	no	yes	no
[57]	High Density	96	11	$Gr_{5}$	1/1	60.00	-	S	no	yes	no
[58]	MYO	8 *	15	$Gr_{6}$	1/1	91.47	-	S	no	yes	yes
[59]	MYO	8 *	6	$Gr_{1}$	10/10	94.70	-	S	no	yes	yes
[63]	MYO	8 *	5	$Gr_{2}$	40/40	92.40	-	G	no	yes	yes
**S-HGR ****	MYO	8 *	5	$Gr_{1}$	306/306	94.96	94.20	S	yes	yes	yes
**G-HGR ****	MYO	8 *	5	$Gr_{1}$	306/306 ***	81.22	80.31	G	yes	yes	yes
**S-HGR ****	MYO	8*	5	$Gr_{1}$	306/306	94.96	94.20	S	yes	yes	yes

${}^{*}$ Myo bracelet used, ${}^{**}$ Specific and General Proposed Models, user-specific (S), and user-general (G) HGR models. ${}^{***}$ Training users are different from testing users; a description of the gestures $Gr_{1}$ to $Gr_{7}$ that the analyzed papers study can be found in the Appendix D.

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
