# Peer review of "An Energy-Based Method for Orientation Correction of EMG Bracelet Sensors in Hand Gesture Recognition Systems"

_sensors, 2020, doi:10.3390/s20216327_

Round 1

Reviewer 1 Report

This work present an algorithm to correct orientation issue of the Myo bracelet for hand gesture recognition.  The algorithm is based on finding the maximum energy channel and then the sensor pods information can be realigned.  By using the proposed classification and orientation correction algorithm, the overall recognition performance has been significantly improved.  I consider this work is ready to be published and would be a great contribution to Sensors based on the following reasons:

  • The overall organization of this work and section structures are consistent with the claimed contributions.
  • The authors provided sufficient background information of related research and studies
  • In section 2, the authors explained in detail of how the system was built and how the data was processed.
  • Testing results are presented clearly.
  • The authors also discussed about limitations of the current work and potential method to address them in future steps.

Author Response

The responses to the reviewer are included in the pdf attached to this message.

Reviewer 2 Report

The authors propose an orientation correction algorithm for hand gesture recognition (HGR) systems using electromyography (EMG) bracelet sensors.

The paper contains the following problems:

  • The authors used the SVM classification method for the hand gesture classification, and they have provided only a brief explanation of the SVM algorithm in section 2.4. The provided explanation must be expanded and followed by corresponding pseudocode describing SVM.
  • The authors in section2.4 need to add an appropriate citation about the SVM algorithm.
  • The authors must justify in section 2.4 why they have used SVM over other alternative options (e.g., deep neural networks).
  • The data in confusion matrices at Tables 1-9 are difficult to read. The authors need to enlarge the tables and the font size of the data.
  • A table summarizing the parameters used for the experiments must be added in section 3, including the regularization parameter C and the kernel parameters for SVM. This table would greatly help the reader to reproduce the experimental results.
  • Each figure, after the title, must be accompanied by a small paragraph containing an explanation.
  • The discussion section must be written without using bullets and the sentence “During the development of this work, several lessons were learned:” must be removed.
  • Finally, the paper language needs polishing since it contains syntax errors.

Based on the above, critical information is missing, and the manuscript should be reevaluated after undergoing a major revision.

Author Response

(The authors gave the same response as above.)

Reviewer 3 Report

The authors propose an interesting method to correct the orientation rotation of the Myo bracelet sensor in a hand gesture recognition system, both in a user-specific and user-general model.

The paper is overall well written and presents, in an accurate way, promising results. They corroborate the necessity of using this kind of algorithm in the field of the hand gesture recognition systems that are based on EMG bracelets like MYO armband.

Another strength of the paper lies in the adoption of two different metrics to quantify the system "goodness": the accuracy and the recognition. This kind of system quality analysis gives a good reference to the reader.

Moreover, the paper is well supported by dedicated appendixes, which supports many of the empirical data provided in the main manuscript.

The references section is overall satisfying, but it can be improved: 14% of the references are from self-citations, while 34% are dated (before 2016).

Nevertheless, there are some issues that must be addressed before possible publication. They are point-by-point listed below.

> In the abstract the distinction between recognition and accuracy parameters is not introduced, thus, it does not appear clear why they provide two different values. It is a common practice to think about them as the same metric.

> In the abstract section is also not clear from which background you extract the values before and after the application of correction. A statement that explains in a few rows what the outcomes refer to should be added.

> There should be some improperly reported statements in the manuscript, that say "any position" concerning the MYO armband placement (see row 144). Did the authors mean any angles? Talking of position in EMG differential measures means also the longitudinal displacement of the armband. It leads to a lot of other problematics such as the proximity to innervation or tendon zones, different behaviors of EMG signals across innervation zones, and so on. The authors should correct the phrase: any position when they refer to angles.

> Row 141: the "waveOut" gesture has not yet introduced. The authors should remain generic.

> Row 215-216: Is the label Y the same values along the third dimension of the MSK matrix? Or is it a generic label for the entire block of EMG data evaluated along with the 7 sliding windows? 

> Equations (6) and (15) are not expressed with mathematical rigor. It is not clear which operator is used for the purpose.

> Which is the rationale behind the choice of an SVM in this kind of application, rather than - for instance- a KNN? It doesn't seem that there is a problem with small datasets.

> Concerning Sec. 2.2: the thresholds have been based on the waveOut gesture. Nevertheless, waveOut asks for a different muscles firing patter, rather than - for example- the waveIn one. Let's think of the flexors' involvement in both the situations: muscular activity which is under the threshold in a gesture is above in an antagonist movement.

> Row 313: The statement: "This matrix is composed of negative scores, and the SVM gives as selected label the one nearest to zero." should be clarified.

> Severe:  The datasets' links are unavailable on 25/09/2020. It makes the repetition of the computations impossible.

> Since the datasets are from previous work, and are not available as per the previous comment, it is not clear how the authors formally "rotate" the MYO bracelet for the purpose of this paper. The authors extracted the same results for other related papers? Datasets from previous works include all the experiments in Fig. 7?

Author Response

(The authors gave the same response as above.)

Round 2

Reviewer 3 Report

The authors present a revised version of a paper that, as previously stated, presents an interesting method to correct the orientation rotation of the Myo bracelet sensor in a hand gesture recognition system, both in a user-specific and user general model.

The paper has been deeply revised and it is, overall, well written and presented. Most of the concerns have been properly addressed, such as formulas, mathematical rigor of eq. (6) and (15), etc.

Most of the choices (e.g., classifier) have been better justified and the pseudocode, added to improve the readability of the score validation routine, surely improved the quality of the paper.

Nevertheless, there are some manuscript parts that are still unclear/improvable.

Below a point-by-point list:

> The authors added 25 new references from the last 4 years, but they didn't use them to improve the related works section for comparison purpose; they simply use them to list possible applications of HGR systems.

> Despite the obvious difficulties in condensate these concepts in a short abstract, the authors should consider to furtherly revise the abstract, because it is still impossible to deduce from this section the differences among the two metrics: recognition and accuracy.

Typically, in the classification context, they are used -independently- to define the same metric. If it is not the case, as in this application, a statement that clarifies the differences should be included.

> In the abstract the authors say that the method increases performance of 38.8% (recognition) and so on. But, which is the "ground truth" reference that has been improved?

Author Response

The responses to the Reviewer and to the Editor comments are included in the document attached to this message.
